# Comparative Analysis of Urso- and Tauroursodeoxycholic Acid Neuroprotective Effects on Retinal Degeneration Models

**DOI:** 10.3390/ph15030334

**Published:** 2022-03-09

**Authors:** Alejandra Daruich, Emilie Picard, Justine Guégan, Thara Jaworski, Léa Parenti, Kimberley Delaunay, Marie-Christine Naud, Marianne Berdugo, Jeffrey H. Boatright, Francine Behar-Cohen

**Affiliations:** 1From Physiopathology of Ocular Diseases to Clinical Development, Centre de Recherche des Cordeliers, Sorbonne University, Paris University, Inserm, F-75006 Paris, France; picardemilie@gmail.com (E.P.); tharajwk@gmail.com (T.J.); lea_parenti@hotmail.com (L.P.); kimberley.delaunay@live.fr (K.D.); marie-christine.naud@crc.jussieu.fr (M.-C.N.); marianne.berdugo@inserm.eu (M.B.); francine.behar@gmail.com (F.B.-C.); 2Ophthalmology Department, Necker-Enfants Malades University Hospital, Assistance Publique Hôpitaux de Paris, Paris University, F-75015 Paris, France; 3Institut du Cerveau (ICM), INSERM, CNRS, AP-HP, Sorbonne University, Pitié-Salpêtrière University Hospital, F-75013 Paris, France; justine.guegan@icm-institute.org; 4Department of Ophthalmology, Emory University School of Medicine, Atlanta, GA 30322, USA; jeffboatright@emory.edu; 5Center of Excellence, Atlanta Veterans Administration Medical Center, Decatur, GA 30033, USA; 6Ophtalmopole, Cochin Hospital, Assistance Publique Hôpitaux de Paris, Paris University, F-75015 Paris, France

**Keywords:** UDCA, TUDCA, retina, retinal detachment, retinal degeneration, neuroprotection

## Abstract

Ursodeoxycholic (UDCA) and tauroursodeoxycholic (TUDCA) acids have shown neuroprotective properties in neurodegenerative diseases, but differential effects of the two bile acids have been poorly explored. The aim of this study was to evaluate the neuroprotective effects of UDCA versus TUDCA in a neuroretinal degeneration model and to compare transcriptionally regulated pathways. The WERI-Rb-1 human cone-like cell line and retinal explants were exposed to albumin and TUDCA or UDCA. Viability, cell death, and microglial activation were quantified. Transcriptionally regulated pathways were analyzed after RNA sequencing using the edgeR bioconductor package. Pre-treatment of cone-like cells with UDCA or TUDCA significantly protected cells from albumin toxicity. On retinal explants, either bile acid reduced apoptosis, necroptosis, and microglia activation at 6 h. TUDCA induced the regulation of 463 genes, whilst 31 genes were regulated by UDCA. Only nineteen common genes were regulated by both bile acids, mainly involved in iron control, cell death, oxidative stress, and cell metabolism. As compared to UDCA, TUDCA up-regulated genes involved in endoplasmic reticulum stress pathways and down-regulated genes involved in axonal and neuronal development. Either bile acid protected against albumin-induced cell loss. However, TUDCA regulated substantially more neuroprotective genes than UDCA.

## 1. Introduction

Primary bile acids (BA) are synthesized from cholesterol in the liver and then excreted into the intestine, where the gut microbiota converts primary BA into secondary BA through chemical modifications. The main function of BA is the emulsification, absorption, and digestion of lipids. However, the hydrophilic secondary BAs, ursodeoxycholic acid (UDCA) and tauroursodeoxycholic acid (TUDCA), the taurine conjugate of UDCA, that are also circulating at low levels, have also shown interesting neuroprotective effects in various neurodegenerative [1] and retinal disease [2,3]. Although antiapoptotic, anti-inflammatory, and antioxidant effects have been demonstrated for these molecules, little is known about primary signaling pathways through which bile acids act as neuroprotectants [4,5,6], and whether both molecules regulate similar pathways is unknown.

A recent systematic review [1] reported that while UDCA reduces apoptosis, reactive oxygen species (ROS) and tumor necrosis factor (TNF)-α production in neurodegenerative models, and reduces nitric oxide (NO) and interleukin (IL)-1β production in neuropsychiatric models, TUDCA reduces ROS and IL-1β production in neurodegenerative models, and decreases apoptosis and TNF-α production, and increases glutathione production in neuropsychiatric models. Both bile acids showed beneficial effects in models of Huntington’s disease, Parkinson’s disease, and Alzheimer’s disease, but the two molecules have not been compared in neurodegenerative models.

In retinal degeneration models, TUDCA inhibited apoptosis and promoted cell survival and function. It protected from caspase-dependent [7,8] and independent (AIF) apoptosis [9] and from endoplasmic reticulum (ER) stress-mediated apoptosis [10,11]. Anti-inflammatory and antioxidant effects have been reported for TUDCA in photoreceptor degeneration models [8,9,12]. Additionally, TUDCA showed beneficial effects on axonal regeneration and amyloid-beta synaptic toxicity in a spinal cord injury model and in cortical neurons [12,13]. The neuroprotective effect of UDCA also has been demonstrated, although in a limited number of animal models [14,15]. The comparative efficacy of TUDCA and UDCA has been only explored in a laser-induced choroidal neovascularization model, showing similar effects on the neovascular complex leakage but probably by different mechanisms [16].

UDCA is approved by regulatory agencies for the treatment of gallstones. The ocular bioavailability of commercially available oral UDCA was recently evaluated after being administered to patients before retinal detachment surgery [3]. In ocular media, the levels of UDCA correlated with the extent of blood–retinal barrier disruption and reached neuroprotective levels. Indeed, UDCA at concentrations measured in subretinal fluid protected photoreceptors from apoptosis and necrosis in rat retinal explants, an ex vivo model of retinal degeneration, and the subretinal fluid of patients who received oral UDCA showed neuroprotective effects in the same model [3]. This study suggested that oral administration of commercial formulations of UDCA could be used to prevent photoreceptor cell loss in patients undergoing an operation for retinal detachment with macula involvement, the main risk factor of incomplete visual recovery despite successful surgery.

Cone photoreceptor cells are densely packed in the fovea and are responsible for visual acuity and color perception. Because the eye is protected by ocular barriers (similar to the brain–blood barrier), the transport of albumin is highly regulated and albumin is not detected in the normal neural retina outside of blood vessels [17]. When ocular barriers are disrupted by pathological processes (retinal detachment, inflammation, glaucoma, trauma, or diabetes), the albumin concentration increases inside the ocular media and tissues [3,18,19] and could contribute to photoreceptor cell death, as shown in neurons [20]. Indeed, we showed that albumin reduced cone viability in a dose-dependent manner [14]. Albumin exposure of retinal cultures was thus used in the current study to mimic pathological conditions that mimic human pathology and that are likely to induce photoreceptor toxicity. Such cultures were treated with UDCA or TUDCA to evaluate their effects on albumin-induced retinal cell death.

Thus, the aim of the current study was to compare the neuroprotective effects of UDCA and TUDCA on albumin-induced retinal cell death and to explore common and differential transcriptional regulated pathways.

## 2. Results

### 2.1. UDCA and TUDCA Protect against In Vitro Albumin-Induced Cone Photoreceptors Death

Incubation with albumin (20 mg/mL) significantly reduced the number of viable WERI-Rb-1 cells (*p* = 0.003), a human cone-like cell line (Figure 1A). This dose of albumin was chosen following previous experiments and measurement of albumin concentrations in the subretinal fluid of patients with retinal detachment, in order to mimic a realistic clinical condition [3]. Pre-treatment of WERI-Rb-1 cells with either TUDCA (1 µM) prevented cell loss (*p* = 0.003), with significance only for TUDCA (*p* = 0.08 for albumin versus UDCA and *p* = 0.0002 for albumin versus TUDCA; Figure 1A), whilst no difference was measured on the neuroprotective effects of UDCA vs. TUDCA (*p* = 0.18) (Figure 1A). LDH release was increased by albumin (*p* = 0.03). Treatment by UDCA or TUDCA decreased LDH release (*p* = 0.003), with only significance for UDCA (*p* = 0.01) No significant difference was seen between UDCA and TUDCA treatment effects (*p* > 0.9) (Figure 1B).

### 2.2. UDCA and TUDCA Protect against Albumin-Induced Cell Death on Ex Vivo Retina Organoculture

Rat neuroretinas, placed on membranes with photoreceptors facing up and exposed to albumin, mimic relevant pathological features of retinal diseases associated with blood–retinal barrier dysfunction and photoreceptor cell death [14,18]. Rat neuroretinas were exposed to albumin or to albumin plus UDCA or TUDCA (10 ng/mL) and cultured for 6 h (Figure 2). Receptor-interacting protein (RIP), a marker of necrotic cell death, was lower in retinas treated with UDCA or TUDCA, with only statistical significance for TUDCA (*p* = 0.01). No significant difference was seen between UDCA and TUDCA treatment outcomes (*p* = 0.5) (Figure 2A). The ratio of cleaved to uncleaved caspase 3, a measure of apoptotic cell death, was lower in retinas treated with UDCA or TUDCA, with only statistical significance for UDCA (*p* = 0.02) (Figure 2B). No significant difference was seen between UDCA and TUDCA (*p* ≥ 0.99) (Figure 2B). The number of photoreceptor cells that were TUNEL positive was reduced in retinas treated with UDCA or TUDCA treatment, with only statistical significance for TUDCA (*p* = 0.03) (Figure 2C). No significant difference was seen between UDCA and TUDCA (*p* = 0.7). Activation of microglial cells represented by the ratio of round/ramified ionized calcium-binding adapter molecule (IBA1)–positive cells, was lower in UDCA and TUDCA-treated retinas as compared to retina exposed to albumin alone (*p* < 0.001), with statistical significance only for TUDCA (*p* = 0.001). No significant differences were seen between UDCA and TUDCA (*p* = 0.2) (Figure 2D).

### 2.3. UDCA and TUDCA Induced Differential Transcriptional Regulation Pathways

The transcriptional regulation pathways induced by UDCA and TUDCA were studied on rat neuroretinas exposed to albumin or to albumin and UDCA or TUDCA (10 ng/mL) for 6 h, using RNA sequencing. TUDCA versus albumin differentially regulated 463 genes (196 up-regulated and 267 down-regulated. On the other hand, UDCA vs. albumin differentially regulated specifically only 31 genes (22 up-regulated and 9 down-regulated) (Appendix A).

Only 19 common genes were significantly differentially regulated by both UDCA versus albumin and TUDCA versus albumin (log2 FC > 0.5, *p* < 0.05). *Mt2A*, *Hmox*, *Atf3*, *Il4i1*, *Mthfd2*, *Gadd45b*, *Myc*, *Gdf15*, *Cebpb*, *Ciart*, *Chac1*, *Asns*, *Fosl1*, *Ppp1r15a*, *Chka*, and *Il27* were up-regulated and *Acta2*, *Rnf144b*, and *Etv5* were down-regulated (Figure 3). Functional enrichment analysis was performed using the Enrichr tool on the Gene Ontology Biological Process database (https://maayanlab.cloud/Enrichr/, accessed on 28 January 2022) to identify 10 main gene pathways that have been enriched in the expression data. Hmox1 and Mt2A involved in transition iron homeostasis are implicated in anti-oxidant processes. Several genes encode transcription factors (*Myc*, *Atf3*, *Etv5*, *Ciart*, *FosL1*, *Cebpb*). Various genes are involved in apoptotic process regulation (*Etv5*, *Asns, Ppp1r15a*, *Chka*, *Myc*, *Hmox1*, *Atf3*). Some genes encode proteins involved in cellular metabolism (*Asns*, *Cebpb*, *Chak*, *Mthfd2*, *Gdf15*, *IL4i1*). Anti-inflammatory mechanisms result from a decrease in *IL27* and an increase in *Il4i1* (Figure 3).

The main pathways differentially regulated by UDCA versus TUDCA are shown in Figure 4. Compared to UDCA, TUDCA up-regulates genes mainly involved in endoplasmic reticulum stress, translation, splicing, and DNA replication. Interestingly, pathways involved in neuronal development, axonogenesis, axon guidance, positive regulation of cell growth, and cell morphogenesis were down-regulated by TUDCA as compared to UDCA, at this early time point after stress exposure (Figure 4). 

Finally, a network of the main genes up-regulated and down-regulated by TUDCA versus control and the involved Hallmark pathways are shown in Appendix A.

## 3. Discussion

In this study, we bring evidence that both UDCA and TUDCA protected the retina from albumin-induced cell death, including apoptosis and necroptosis, in a model of retinal detachment, confirming results previously published using oral UDCA in retinal detachment [3]. Both bile acids protected human cones from albumin-induced cell death, with variable effects depending on the apoptosis or necroptosis analyzed read-out, which may be due to a specific timing of the analysis, knowing that we are analyzing dynamic processes. Further studies using longitudinal analysis of cell death with vital probes are being conducted to better analyze the kinetics of events. Depending on the cell death pathways and the measured parameters, one or the other drug was more efficient but both drugs also protected the whole retina.

Following RNA-seq, a limited number of common genes were found to be regulated by both TUDCA and UDCA at the time point assessed. They could eventually be more specifically linked to bile acid receptors and signaling pathways. Interestingly, the up-regulated genes encode proteins known to play major roles in the protection of the retina against oxidative stress and against ferroptosis. Among those common genes, Mt2A encodes a metallothionein protein that binds divalent heavy metal ions, altering the intracellular concentration of heavy metals in the cell. It acts as an anti-oxidant, anti-apoptosis, detoxification, and anti-inflammatory enzyme and protects against hydroxyl free radicals [18]. Metallothionein was shown to protect retinal pigment epithelial cells against apoptosis [19] and to play a major role in protecting the eye from oxidative stress [21]. Hmox1 is a component of a protection mechanism against iron overload and a pivotal player in oxidative stress with differential effects depending on the timing of expression and studied models [22], whilst ATF3 is a transcription factor involved in ferroptosis modulation [23]. In the retina, ATF3 protected against ganglion cell death in crush models [24] and was identified in survival pathways in the rd10 retinal degeneration mouse model [25]. Chac1 is a glutathione-degrading enzyme induced by ferroptosis. Interleukin-4-induced-1 (IL4i1) is an amino acid oxidase involved in arginine metabolism. Recently, IL4i1 was shown to elicit a cell-protective gene expression program inhibiting ferroptosis through the generation of indole-3-pyruvate (I3P) from tryptophan [26] and to induce macrophage M2 [27]. Altogether, these regulations indicate that bile acids could also protect the retina from ferroptosis, a recently discovered cell death mechanism involved in retinal detachment, inherited retinal dystrophies and age-related macular degeneration [28]. In the brain, GADD45, expressed in neurons in response to a wide range of stimuli, protects against apoptosis and induces the expression of the anti-apoptotic protein Bcl-2 [29]. The bifunctional methylenetetrahydrofolate dehydrogenase/cyclohydrolase (MTHFD2) is a mitochondrial one-carbon enzyme central to folate-mediated one-carbon metabolism in mitochondria and directly linked with the serine/glycine metabolism, which was found to be crucial for retinal health [30]. The limited number of genes regulated early by both UDCA and TUDCA play major roles in the control of cell death and metabolism and in the regulation of retinal inflammation.

Since our analysis has been performed at 6 h after treatment, we explored only the early gene regulations. Transcription factors were up-regulated, suggesting that other regulations could occur at later time points.

Whether TUDCA and UDCA could exert synergistic effects should be explored. Indeed, TUDCA regulated the expression of a large number of genes that are not regulated by UDCA. Such genes encode proteins that belong to oxidative stress defense such as peroxiredoxin 1, 2, 4, 6 known to play important roles in the retina, particularly peroxiredoxin 6 [31] and SOD1. Of note, the number of genes encoding proteins implicated in synaptic transmission and in sensory organ development is down-regulated by TUDCA specifically, which could represent potential regulation of neuronal connectivity and organization, which is in line with the beneficial effects of TUDCA that was reported in neurodegenerative diseases of the central nervous system. It cannot be excluded that additional TUDCA regulations and potential protective effects result from the effect of taurine itself as taurine is known to prevent retinal neurodegeneration [32,33]. Photoreceptors are particularly rich in taurine, and all retinal cells take up taurine from the extracellular milieu. High- and low-affinity taurine transporters have been described in the retina [34] but a taurine-specific receptor has not been yet identified. However, whether taurine and TUDCA share the same receptor remains to be elucidated [35].

In this experiment, TUDCA was more efficacious than UDCA in reducing microglial activation. One study reported the comparative efficacy of TUDCA and UDCA in a laser-induced choroidal neovascularization (CNV) model, showing similar effects in terms of CNV suppression [13]. However, only TUDCA was associated with early inhibition of VEGF in the retina after laser injury, which indicates that other mechanisms might be involved in the suppression of CNV by UDCA. Accordingly, we have shown here that various pathways are differentially regulated by both acids.

Although receptors and transporters for bile acids have been partially found in the retina, specific interactions of UDCA and TUDCA with their receptors or transporters have been rarely explored. The farsenoid receptor (FXR) poorly binds UDCA and TUDCA and is therefore unlikely to mediate the signaling pathway in the retina where the receptor does not seem to be expressed [36]. Other bile acid receptors such as the vitamin D receptor, pregnane X receptor, gluco- and mineralocorticoid receptors, membrane receptors TGR5, sphingosine 1-phosphate receptor 2 (SIP1 and 2), and α5β1 integrin are expressed in the retina and could mediate these effects [2]. The interaction between TUDCA and TGR5 was shown in retinal ganglion cells [2]. In addition, it has been shown that TUDCA could activate the MerTK receptor in RPE cells [37] and could directly interact with rhodopsin [34]. No receptor interaction has been described for UDCA in the retina. Therefore, the signaling mechanisms leading to the regulation of gene expression by each compound, common to both bile acids, remain to be specifically studied.

We acknowledge some weaknesses in this study. The fact that the transcriptomic analysis was performed at a single time point does not take into account the natural kinetics of events and rather gives a snapshot image. From the transcriptomic regulations, we identified that ferroptosis could be a target cell death mechanism of bile acids, but we have not yet confirmed this finding.

## 4. Materials and Methods

### 4.1. Cell Line and In Vitro Model

Treatment with Albumin at 20 mg/mL has been shown to decrease WERI-Rb-1 human cone cell line (HTB-169, ATCC, Manassas, VA, USA) viability by about 40% [3] The WERI-Rb-1 human cone cell line was cultured on Roswell Park Memorial Institute (RPMI)-1640 medium (Thermo Fisher Scientific) and supplemented with 10% fetal bovine serum and 1% penicillin-streptomycin. Cells were then treated by 1 µM of UDCA (Sigma Aldrich, *n* = 4–8/group) or TUDCA (Sigma Aldrich, *n* = 4–8/group) 1 h earlier to add albumin. After 24 h of culture, 100 µL of incubation medium was collected to determine the cytotoxicity by lactate dehydrogenase (LDH) release (Sigma Aldrich Chemical Co., Saint-Louis, MO, USA), as previously described [3]. CellTiter (CellTiter 96^®^ AQueous One Cell Proliferation Assay Solution, Promega Corporation, Charbonnières-Les-Bains, France) was used to assess mitochondrial activity. It was added to the cells and then incubated for 2 h. The absorbance was read at 492 nm. The results were calculated on percentage, by reporting the absorbance of cells by group to the mean of the control group.

### 4.2. Animals and Ex Vivo Model of RD

Adult male Wistar rats (Janvier Labs, Le Genest St Isle, France, *n* = 8) were fed with a standard laboratory diet and ad libitum tap water in a room maintained at 21° to 23 °C with a 12 h light/12 h dark cycle (6 a.m. to 6 p.m.) for 7 days, before being sacrificed by carbon dioxide inhalation, following French and European legislation (Décret n°2013-118 du 1^er^ Février 2013).

Retinal explants were created to mimic retinal degeneration conditions. Neuroretinas were dissected from freshly enucleated eyes, separated from the retinal pigment epithelium, divided into two parts, and then transferred to 0.2 mm polycarbonate membranes (Millipore, Saint Quentin En Yvelines, France) with the photoreceptor layer facing up [3]. The membranes were next placed into a 6-well culture plate containing Dulbecco’s Modified Eagle’s Medium (Thermo Fisher Scientific, Waltham, MA, USA) and 3% fetal bovine serum (3.9 mL/well). Retinal explants were then treated with 100 µL of albumin alone (bovine serum albumin at 12 mg/mL, control) or containing UDCA or TUDCA at 10 ng/mL (*n* = 5/group). Concentrations of albumin, UDCA, and TUDCA were based on previous experiences [3].

The dilutions of UDCA, TUDCA, or vehicle were successively performed in ethanol 100% (10 mg/mL), phosphate-buffered saline (PBS), and medium (the same preparation for the culture wells). Albumin was added at the last dilution. Treated explants were cultured for 6 h. Immunohistochemistry, Western blotting, and ribonucleic acid (RNA) sequencing were performed on explants, as described below.

### 4.3. Western Immunoblotting Analysis

Explants (*n* = 5 per group) were lysed with MPER buffer (Thermo Fisher Scientific) and then centrifuged at 13,000× *g* for 5 min at 4 °C. Protein concentrations were calculated using the Micro BCA protein assay (Thermo Fisher Scientific, Waltham, MA, USA). Five to 10 mg of total extract was then mixed with protein loading buffer (Thermo Fisher Scientific, Waltham, MA, USA), as per the manufacturer’s instructions. Samples were loaded onto a 4 to 12% bis-tris gel (Thermo Fisher Scientific, Waltham, MA, USA) and proteins were transferred onto nitrocellulose membranes. Nonspecific binding was blocked with 5% nonfat dry milk in Tween/tris-buffered saline, then membranes were incubated overnight at 4 °C, with the primary antibody against Caspase 3 (1:500; Clone C92-605, BD Transduction Laboratories, CliniSciences, Nanterre, France) and receptor-interacting protein (RIP) kinase 1 (1:500; Clone 38, BD Transduction Laboratories, Franklin Lakes, NJ, USA) or actin (1:4000; Sigma-Aldrich, Saint-Louis, MO, USA), followed by incubation with the supplier-recommended dilution of horseradish peroxidase-conjugated secondary antibody for 1 h (Vector Laboratories, Eurobio, Les Ulis, France). Protein bands were visualized by an enhanced chemiluminescence reaction (Thermo Fisher Scientific, Waltham, MA, USA) using a Bioimaging system (Invitrogen iBright Imaging Systems, Thermo Fisher Scientific, Waltham, MA, USA). The gray values of specific bands were quantified using ImageJ, and the protein signals of interest were reported relative to the actin signal or to their respective cleaved forms (for Caspase 3) or actin (for RIP).

### 4.4. Immunohistochemistry and Fluorescence Intensity Evaluation

Explants were rinsed in 1× phosphate-buffered saline, fixed for 20 min with 4% paraformaldehyde (Inland Europe, Conflans sur Lanterne, France), infiltrated in a sucrose gradient series, and then mounted in Tissue-Tek optimum cutting temperature (OCT) (Siemens Medical, Puteaux, France). Immunohistochemistry was performed on 10 µm-thick sections. Microglia and macrophages were immunodetected with ionized calcium-binding adapter molecule 1 (IBA1) (Wako Pure Chemical Industries, Neuss, Germany) staining. The terminal deoxynucleotidyl transferase-mediated biotinylated UTP nick end-labeling (TUNEL) reaction was also performed. The protocol was adapted from the manufacturer’s protocol (Roche Diagnostics, Meylan, France). Sections were counterstained with 4,6-diamidino-2-phenylindole (DAPI, Sigma-Aldrich, Saint-Louis, MO, USA). The sections were photographed with a fluorescence microscope (BX51, Olympus, Rungis, France), using identical exposure parameters for all compared samples. Blind quantifications were realized on photographs acquired at 20× magnification with ImageJ software. IBA1-positive cells were quantified based on their differential shapes (round amoeboid or ramified dendritic form), as previously reported [38], and the results were expressed as a ratio (mean of 8 photographs/explant, *n* = 5 explant/group). TUNEL-positive cells were quantified on 4 photographs/explant, *n* = 5/group).

### 4.5. Ribonucleic Acid (RNA) Sequencing and RNA-Seq Data Analysis

Retinas (*n* = 3 per group) were frozen immediately after isolation. Total RNA was extracted using a Precellys homogenizer (Bertin, Montigny-le-Bretonneux, France) and an RNeasy Mini Kit (Qiagen, Hilden, Germany). The quantity and quality of the total RNAs extracted were assessed by the Tapestation 2200 (Agilent Technologies, Santa Clara, CA, USA). cDNA libraries were prepared using a stranded mRNA polyA selection (KAPA hyper mRNA kit, Roche Diagnostics, Meylan, France) and sequenced with the Illumina Novaseq 6000 sequencing system. For each sample, we performed 34 million paired-end reads, 150 bases in length (NovaSeq 6000 SP Reagent Kit, 300 cycles, 200 Gbases).

The quality of raw data was evaluated with FastQC. Poor quality sequences were trimmed or removed with Trimmomatic software, in order to retain only good quality paired reads. Star v2.5.3a was used to align reads on the rn6 reference genome using standard options. Quantification of gene and isoform abundances was performed with rsem 1.2.28. Finally, normalization and differential analysis were conducted with the edgeR bioconductor package. Multiple hypothesis-adjusted *p*-values were calculated with the Benjamini–Hochberg procedure to control the FDR. The Log2 fold-change threshold was set to 0.5 and the FDR threshold to 0.05. A pre-filtering was applied with these parameters: at least 30% of samples per group respect a minimum CPM threshold of 1. Enrichment analyses were performed with GSEA (Gene Set Enrichment analysis) on MsigDB collection gene sets with the clusterprofiler R package.

### 4.6. Statistics

Results are presented as the mean ± SEM. Analyses were performed using GraphPad (version 9.1.0, GraphPad Software, La Jolla, CA, USA). Multiple comparisons were performed using the Kruskal–Wallis test followed by Dunn’s multiple comparisons post test. *p*-values < 0.05 were considered statistically significant.

## 5. Conclusions

In conclusion, both TUDCA and UDCA could protect against retinal cell loss in ex vivo models of retinal detachment. Bile acids protected from apoptosis and necroptosis and could also potentially protect against ferroptosis, according to the genes regulated by both drugs. This will be a subject of further study. The early transcriptomic regulation induced by both molecules suggests that bile acids regulate mitochondrial and amino acid metabolism, control oxidative stress, and IL27 and IL4i1-mediated neuronal inflammation. They also activate transcription factors that could induce subsequent regulation, which should be studied by transcriptomic analysis at additional time points. TUDCA specifically regulates a much larger number of genes and pathways involved in neurogenesis and oxidative stress than UDCA. It cannot be excluded that these additional regulatory effects could be related to the taurine conjugate, known to exert protective effects on numerous models of retinal degeneration. Bile acid signaling in the retina still remains poorly understood and should be studied in depth taking into account the increasing links suspected between retinal diseases and microbiota [39]. 

## Figures and Tables

**Figure 1 pharmaceuticals-15-00334-f001:**
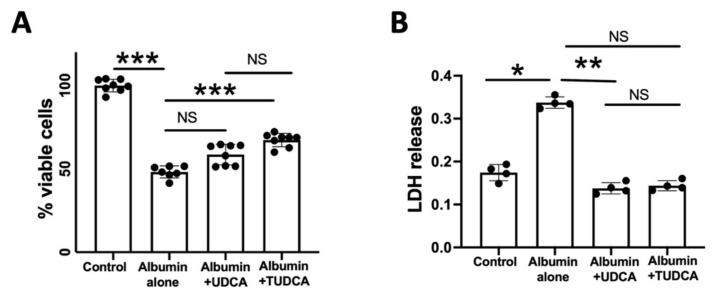
**UDCA and TUDCA protect from albumin-induced cell death in vitro.** (**A**) The % of viable cones cells decreased with albumin (20 mg/mL) but was higher when cones received 1 µM of UDCA or TUDCA, with only significance for TUDCA. No significantly difference was seen between UDCA and TUDCA. (**B**) LDH release was increased by albumin. Treatment by UDCA or TUDCA decreased LDH release, with only significance for UDCA. No significant difference was seen between UDCA and TUDCA treatment outcomes. The results are presented as the means ± SEM, *n* = 4–8, Kruskal–Wallis and Dunn’s multiple comparisons post hoc test (* *p* < 0.05; ** *p* < 0.01, *** *p* < 0.001).

**Figure 2 pharmaceuticals-15-00334-f002:**
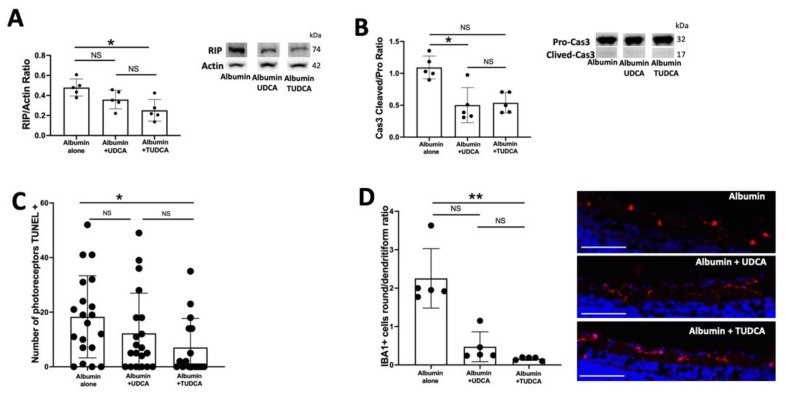
**UDCA protects from albumin-induced cell death ex vivo****.** Rat retina explants were treated by albumin 12 mg/mL or albumin and UDCA or TUDCA 10 ng/mL and cultured 6 h. (**A**) Western blotting quantification. Receptor-interacting protein (RIP)/actin ratio was lower in retinas treated by UDCA or TUDCA, with only significance between for TUDCA vs. albumin. No significant difference was seen between UDCA and TUDCA. (**B**) Cleaved/pro-Caspase 3 ratio was lower in retinas treated by UDCA or TUDCA with only significance between for UDCA vs. albumin. No significant difference was seen between UDCA and TUDCA. (**C**) Number of photoreceptors TUNEL + was reduced in retinas treated by UDCA and TUDCA compared to albumin alone, with only significance for TUDCA. No significant difference was seen between UDCA and TUDCA. (**D**) Round/ramified ionized calcium-binding adapter molecule (IBA1)-positive cells (red) ratio was lower in UDCA or TUDCA-treated retinas, with significance only between for TUDCA vs. Albumin. No significant difference was seen between UDCA and TUDCA (*p* = 0.2). Scale bars: 100 µm. The results were presented as the means ± SEM, *n* = 4–8, Kruskal–Wallis and Dunn’s multiple comparisons post hoc test (* *p* < 0.05; ** *p* < 0.01, *p* < 0.001).

**Figure 3 pharmaceuticals-15-00334-f003:**
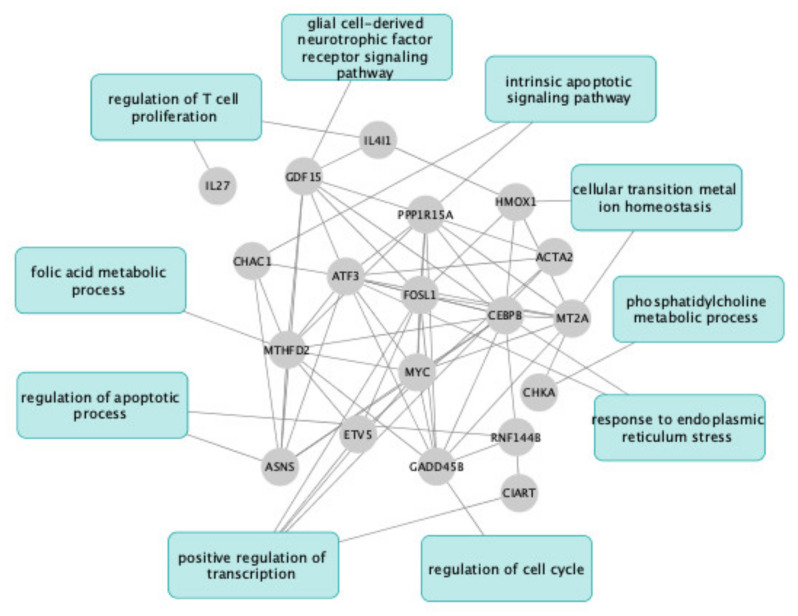
**Nineteen common genes differentially regulated by both UDCA versus albumin and TUDCA versus albumin and gene ontology biological process**. Differential analysis was performed on UDCA vs. albumin and TUDCA vs. albumin. Differentially expressed genes were selected with FDR < 0.05 and logFC > 0.5. 19 genes were in common between these two analyses. A network was constructed with StrinDB to connect those genes. Then Enrichr was used to select the most regulated pathways for those genes. Finally, the graph was created with Cytoscape tool.

**Figure 4 pharmaceuticals-15-00334-f004:**
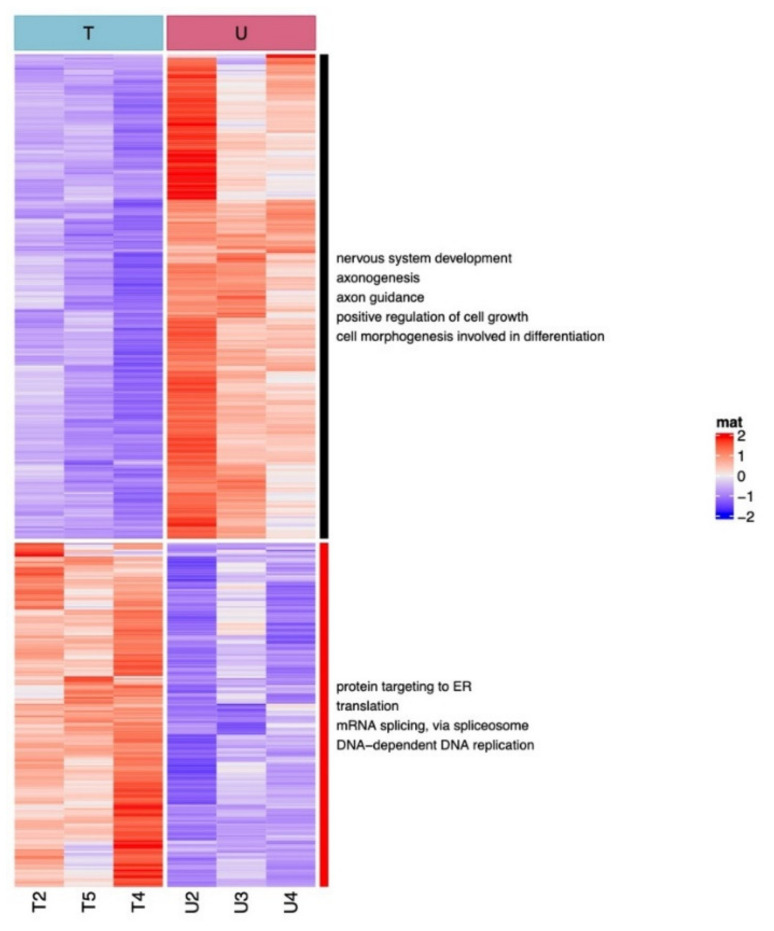
**Genes differentially regulated by TUDCA vs. UDCA and main biological pathway**. Starting with genes deregulated in TUDCA vs. UDCA, a heatmap was constructed with diverging color scale (red for up-regulated genes and blue for down-regulated genes). Genes (in line) were ordered using Euclidean distance. Two major gene clusters appeared. They were annotated with Enrichr.

## Data Availability

Data is contained within the article and Appendix A.

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
