# Peer review of "Comparative Analysis of Urso- and Tauroursodeoxycholic Acid Neuroprotective Effects on Retinal Degeneration Models"

_pharmaceuticals, 2022, doi:10.3390/ph15030334_

Round 1

Reviewer 1 Report

Please label the columns in fig. 1 and 2 clearly - to which group belong the columns 1 - 4 - the +/-0 rows are hard to assign to the columns.

In Abstract, you use the expression "up-regulated genes in endoplasmic stress..." - are those upregulated against e.stress or cause they e. stress - you mean against...

Please correct spacing errors (e.g. line 53).

Author Response

We thank the reviewer 1 for these comments.

Please label the columns in fig. 1 and 2 clearly - to which group belong the columns 1 - 4 - the +/-0 rows are hard to assign to the columns.

We have modified the style of the label on Figures 1 and 2

In Abstract, you use the expression "up-regulated genes in endoplasmic stress..." - are those upregulated against e.stress or cause they e. stress - you mean against...

We have corrected as follows:

“TUDCA up-regulated genes involved in endoplasmic reticulum stress pathways”

Please correct spacing errors (e.g. line 53).

We have corrected the spacing errors

Reviewer 2 Report

The paper "Comparative analysis of urso- and tauroursodeoxycholic acid 2
neuroprotective effects on retinal degeneration models" describes an interesting study reporting the neuroprotective properties of bile acids on retinal degeneration models.

the introduction is very short and not very significant for the study that authors conducted.

ml must be mL.

the reported effects were not very significant although promising.

I think that the molecular target of these effects must be investigated.

Author Response

We thank the reviewer 2 for these comments.

The paper "Comparative analysis of urso- and tauroursodeoxycholic acid 2
neuroprotective effects on retinal degeneration models" describes an interesting study reporting the neuroprotective properties of bile acids on retinal degeneration models.

the introduction is very short and not very significant for the study that authors conducted.

We have improved the introduction and included two paragraphs to extend the context of the study as follows:

Primary bile acids (BA) are synthesized from cholesterol in the liver and then excreted into the intestine, where they are converted by gut microbiota into secondary BA through chemical modifications. The main function of BA is the emulsification, absorption, and digestion of lipids. Interestingly, the hydrophilic secondary BAs, ursodeoxycholic acid (UDCA) and tauroursodeoxycholic acid (TUDCA), the taurine conjugate of UDCA, that are also circulating at low levels in the blood, have  shown neuroprotective effects in various neurodegenerative[1] and retinal disease [2,3]. Although their antiapoptotic, anti-inflammatory, and antioxidant properties have been characterized,  the signaling pathways through which bile acids act as neuroprotectants [4–6] and whether both molecules regulate similar pathways remains have been poorly explored.  

A recent systematic review [1] reported that while UDCA reduces apoptosis, reactive oxygen species (ROS) and tumor necrosis factor (TNF)-α production in neurodegenerative models, and reduces nitric oxide (NO) and interleukin (IL)-1β production in neuropsychiatric models, TUDCA reduces ROS and IL-1β production in neurodegenerative models, and decreases apoptosis and TNF-α production, and increases glutathione production in neuropsychiatric models. Both bile acids showed beneficial effects in models of Huntington's disease, Parkinson’s disease and Alzheimer's disease, but the two molecules have not been compared in models of retinal degeneration.

ml must be mL.

We have corrected.

the reported effects were not very significant although promising.

The statistical significance is not reached for all parameters when using the Kruskal Wallis test followed  by the Dunn’s multiple comparisons post test recommended to compare more than 2 groups. Statistical significance is reached when two groups are compared, due to the number of samples.

Using Mann Withney-test and comparing only albumin vs albumin+ UDCA a significant p value is reached (p=0.002). In figure 1B, the same is shown for TUDCA, that is very similar to UDCA response.

But the comparison of all groups allowed to show slight differences between different readouts, reflecting different interference of the two molecules with different mechanisms of cell death.

I think that the molecular target of these effects must be investigated.

The aim of this study was to compare the effects of UDCA and TUDCA on various cell death markers and to identify the short term transcriptional regulations induced by the two molecules using a non- biased and untargeted method.

The advantage of such analysis is that it identifies without a priori all regulated genes, but the drawback is that it does not explore one specific target in depth.

The results, unexpectedly, shows that although both drugs are able to protect the cones and the whole neural retina  from albumin toxicity, the number of genes regulated by the two BA are very different.

Common genes will be analyzed in depth and particularly the receptor that mediates these regulations will be studied and published separately.

Reviewer 3 Report

The present study has evaluated the neuroprotective role of UDCA in comparison to TUDCA in a neuroretinal degeneration model. The authors have used a commercial cell line but also retina explants.

My questions are:

  • Why Benjamini-Hochberg method was used to control false discovery rate in the RNA-seq experiments?
  • Did the author analyze replicates for each condition in the RNA-seq experiments or a pool was analyzed? How many replicates were used in the RNA-seq experiments?
  • Did the authors study the effect of TUDCA and UDCA alone? Which pathways were altered? Which effect had prolonged time treatments?

Author Response

We thank the reviewer 3 for these comments.

The present study has evaluated the neuroprotective role of UDCA in comparison to TUDCA in a neuroretinal degeneration model. The authors have used a commercial cell line but also retina explants.

My questions are:

  • Why Benjamini-Hochberg method was used to control false discovery rate in the RNA-seq experiments?

Due to the large number of genes in a typical RNA-Seq data set, correction for multiple comparisons is very important. The FDR (Benjamini and Hochberg, 1995) provides an attractive measure of control for multiple testing in genomic settings. This method is proposed as the default procedure to control for false positives in EdgeR R package wich was used for differential analysis.

  • Did the author analyze replicates for each condition in the RNA-seq experiments or a pool was analyzed? How many replicates were used in the RNA-seq experiments?

We did not use a pool, but rather used replicates for each conditions ( 3 retinas  per group) (Line 398-methods section).

  • Did the authors study the effect of TUDCA and UDCA alone? Which pathways were altered? Which effect had prolonged time treatments?

UDCA transcriptional regulations in retinal explants have been already studied and published (PMID: 33537951). The specific transcriptional regulations of TUDCA as compared to control is shown in supplementary Table 1 for DE genes.  A new figure representing main regulated pathways was added to the manuscript as supplementary table 1.

Supplementary Figure 1: Supplementary Figure 1: RNA-seq analysis of rat retinal explants exposed to albumin alone or albumin + TUDCA over 6 hours. Network of the 4 top Hallmark pathways (A) with related genes up-regulated (red) and down-regulated (blue) by TUDCA and the significant involved Hallmark pathways (B).

We have used the albumin-induced retinal degeneration model because it was reproductible and closer to clinical conditions such as retinal detachment where blood-retinal barrier is broken and high protein concentration is found in contact with the retina. We have tested the effects of both bile acids at different time-points in retinal cultures (3h, 6h, 24h and 48h). We found that markers of cell death were higher and more reproductible at at 6hs. We then prefer to use this time point to perform RNA-seq experiments.

We had indicated this limitation in the discussion section as follows:

“We acknowledge some weaknesses in this study. The fact that transcriptomic analysis was performed at a single time point does not take into account the natural kinetics of events and rather gives a snap shot image.”

Round 2

Reviewer 2 Report

the manuscript has been revised accordignly.